# Prospective multicenter study using artificial intelligence to improve dermoscopic melanoma diagnosis in patient care

## Abstract

**Background** Early detection of melanoma, a potentially lethal type of skin cancer with high prevalence worldwide, improves patient prognosis. In retrospective studies, artificial intelligence (AI) has proven to be helpful for enhancing melanoma detection. However, there are few prospective studies confirming these promising results. Existing studies are limited by low sample sizes, too homogenous datasets, or lack of inclusion of rare melanoma subtypes, preventing a fair and thorough evaluation of AI and its generalizability, a crucial aspect for its application in the clinical setting.

**Methods** Therefore, we assessed "All Data are Ext" (ADAE), an established open-source ensemble algorithm for detecting melanomas, by comparing its diagnostic accuracy to that of dermatologists on a prospectively collected, external, heterogeneous test set comprising eight distinct hospitals, four different camera setups, rare melanoma subtypes, and special anatomical sites. We advanced the algorithm with real test-time augmentation (R-TTA, i.e., providing real photographs of lesions taken from multiple angles and averaging the predictions), and evaluated its generalization capabilities.

**Results** Overall, the AI shows higher balanced accuracy than dermatologists (0.798, 95% confidence interval (CI) 0.779–0.814 vs. 0.781, 95% CI 0.760–0.802; $p = 4.0\mathrm{e}{-}145$), obtaining a higher sensitivity (0.921, 95% CI 0.900–0.942 vs. 0.734, 95% CI 0.701–0.770; $p = 3.3\mathrm{e}{-}165$) at the cost of a lower specificity (0.673, 95% CI 0.641–0.702 vs. 0.828, 95% CI 0.804–0.852; $p = 3.3\mathrm{e}{-}165$).

**Conclusion** As the algorithm exhibits a significant performance advantage on our heterogeneous dataset exclusively comprising melanoma-suspicious lesions, AI may offer the potential to support dermatologists, particularly in diagnosing challenging cases.

## Plain language summary

Melanoma is a type of skin cancer that can spread to other parts of the body, often resulting in death. Early detection improves survival rates. Computational tools that use artificial intelligence (AI) can be used to detect melanoma. However, few studies have checked how well the AI works on real-world data obtained from patients. We tested a previously developed AI tool on data obtained from eight different hospitals that used different types of cameras, which also included images taken of rare melanoma types and from a range of different parts of the body. The AI tool was more likely to correctly identify melanoma than dermatologists. This AI tool could be used to help dermatologists diagnose melanoma, particularly those that are difficult for dermatologists to diagnose.

Melanoma, the leading cause of skin cancer deaths worldwide, has increased in incidence over the last few decades[1–3]. Early detection of this disease can reduce the size and extent of surgery as well as adverse effects of late-stage systemic therapies and is thus beneficial for patients, especially if detected pre-metastasizing[4]. Traditionally, physicians diagnose lesions using visual inspections and dermoscopy[5,6]. Prediction quality thereby depends on the expertise and experience of the dermatologist[7]. Considering that the demand for such experts is increasing due to changing epidemiology, ethnographic trends, and skilled advertising[8] and that it is difficult to find such

experts, innovative diagnostic approaches are required, especially for atypical or uncommon cases.

Recently, artificial intelligence (AI) systems for melanoma detection have emerged as a promising tool, with numerous retrospective studies reporting that AI algorithms can match or even surpass the diagnostic accuracy of experienced dermatologists in artificial settings[7,9–11]. While the findings of these retrospective studies are undoubtedly encouraging, hinting at an improvement in patient care while reducing dermatologists' workload, a lack of prospective evaluations remains[12]. Prospective studies typically

✉ e-mail: titus.brinker@dkfz.de

allow for more unbiased, complete, and tailored evaluations, but the few existing prospective analyses of AI-based melanoma detection suffer from limitations such as a single-center design and a relatively small number of lesion samples, particularly with respect to rare melanoma subtypes[13–15].

In this study, we address these limitations, thereby taking a substantial step towards comprehensive, prospective evaluations of AI-based melanoma detection by evaluating the open-source AI algorithm "All Data Are Ext"[16] (ADAE) on a heterogeneous dataset. ADAE is a binary melanoma classifier and ranked first in the Society for Imaging Informatics in Medicine (SIIM) and International Skin Image Collaboration's (ISIC) Challenge 2020[17]. The dataset we introduce is heterogeneous and shows substantial domain diversity (see Fig. 1 for the feature distribution of our dataset). We ensure a broad representation of potential real-life clinical and technical settings by using a multicenter design involving eight German university hospitals as well as four distinct hardware configurations. In addition, we demonstrate a strong performance of the algorithm on difficult-to-diagnose lesions, which suggests integrating AI as a supportive tool for dermatologists in diagnosing particularly challenging cases.

## Methods
### Study design
This prospective, multicenter study was approved by the respective institutional review boards of the Technical University Dresden (BO-EK-53012021), the Friedrich-Alexander University Erlangen-Nuremberg (69_21 Bc), the University Duisburg-Essen (20-9784-BO), the University Hospital Mannheim (2010-318N-MA), the LMU Munich (21-0182), the University Regensburg (20-2190-103), as well as the University Würzburg (293/20_z) and adheres to the Declaration of Helsinki guidelines. Specific IRB approval was not required from the Charité Berlin because the Berufsordnung der Ärztekammer Berlin (professional code of conduct of the medical association Berlin), §15(2), states that additional approval is not necessary for a study across multiple centers if there is approval from another IRB of a German University or medical association. The STARD 2015 reporting standards[18] were followed, and written informed consent was obtained from all participating patients.

Data on clinically suspected melanomas that were subsequently excised, consisting of dermoscopic images and patient-specific metadata (including age, Fitzpatrick skin type, lesion localization, and diameter), were prospectively gathered as part of routine clinical practice from eight university hospitals in Germany (located in Berlin, Dresden, Erlangen, Essen, Mannheim, Munich, Regensburg, and Wuerzburg) between April 2021 and March 2023.

### Participants
Participants for this study were eligible if they met all of the following criteria: at least 18 years of age and presenting with clinically melanoma-suspicious skin lesions. Patients were excluded if these melanoma-suspicious lesions had undergone pre-biopsy procedures, were located near the eye or beneath the fingernails or toenails, or had person-identifying features (such as tattoos) in the immediate vicinity of the lesion due to data privacy concerns.

### Data collection
After informed consent, imaging and dermoscopic examination were performed, and melanoma-suspicious lesions were subsequently excised. All lesions were histopathologically diagnosed by at least one experienced (dermato)pathologist at the respective hospital. During clinical examinations, a dermatologist captured six dermoscopic images of each suspected melanoma lesion while deliberately introducing random variations in the orientation/angle, position, and operational mode of the dermatoscope, including both polarized and nonpolarized settings. A dermatologist hereby is defined as a doctor who studies and treats skin diseases in a Department of Dermatology, but has not necessarily completed board certification yet. To mitigate the influence of potential confounding variables, dermatologists were explicitly instructed to avoid known artifacts (such as skin markings).

All images were acquired using one of four distinct hardware configurations that were consistently used across the participating medical centers (see Supplementary Methods).

### Model training and testing
As we employed the ready-to-use ADAE algorithm for binary classification of lesion images into melanoma and non-melanoma, no additional training was needed. ADAE is trained solely on public data from the respective 2020 and 2019 (which includes 2018 data) SIIM-ISIC challenges, comprising a total of 58,457 lesions, 5106 of which were labeled melanoma[19–22]. The ensemble consists of 18 convolutional neural network (CNN)-models (16 EfficientNets B3-7, 1 SE-ResNext101, 1 ResNest101), of which four additionally incorporate patient meta-data (including sex, age, and lesion location). Since the best model of each 5-fold cross-validation is kept, this totals 90 models in the final ensemble (for a detailed description of the algorithm and its training procedure, please refer to the original paper[16]).

To ensure that the algorithm runs correctly, it was tested on the SIIM-ISIC 2020 data according to a publicly available script on GitHub (https://github.com/ISIC-Research/ADAE/blob/main/predict.py)[23]. The resulting AUROC scores match those available in the literature (ISIC validation: ours 0.945 vs. literature 0.949, ISIC test: 0.951 vs. 0.950)[13,17].

Since our dataset includes six images per lesion, ADAE is adapted with R-TTA to utilize these additional images, as this has proven to be beneficial with respect to diagnostic performance, robustness and uncertainty estimation[24]. There, all six real images are fed to the algorithm at test time, and the resulting outputs are then aggregated to one final prediction. A comparison of the diagnostic accuracy of ADAE with versus without R-TTA is shown in Supplementary Figs. 6 and 7, which also include the scores for the individual models that the ensemble comprises.

To find a suitable threshold differentiating positive from negative predictions, a validation set was split from the data, namely, the data of hospital 8, as it also has its own unique technical domain while being sufficiently large enough to allow for a representative estimate. Therefore, the threshold is set such that a sensitivity of at least 85% is exceeded, as sensitivities of approximately 80–85% are realistic in a clinical setting[9,25,26].

### Statistics and reproducibility
The difference between ADAE and dermatologists' diagnostic accuracy was primarily quantified through balanced accuracy, and secondarily via sensitivity and specificity. For each endpoint, pairwise two-sided Wilcoxon signed-rank tests were used to compare the respective metrics. To evaluate the generalizability of the algorithm on different subsets, the Breslow–Day test for homogeneity of the odds ratio was used[27]. Hypothesis H0 is a constant odds ratio over a stratified variable $k$, indicating whether there is a significant association between prediction and $k$. The algorithm's predictive ability was assessed using the AUROC. Differences in model performance were assessed by statistically comparing the corresponding AUROCs using DeLong's method[28].

To reduce the impact of stochastic events, mean values for each metric were calculated using 1000 bootstrap iterations. The corresponding 95% confidence intervals (CIs) were determined using the nonparametric percentile method. $P$-values smaller than 0.05 were considered statistically significant. Statistical analysis was performed using SciPy 1.11.2[29] and R[30].

### Reporting summary
Further information on research design is available in the Nature Portfolio Reporting Summary linked to this article.

## Results
### Patient characteristics
Our dataset comprises images of a total of 1910 skin lesions that were clinically suspected to be melanoma from 1716 patients at eight German university hospitals collected between April 2021 and March 2023 (for detailed patient characteristics, see Table 1). The patient's age at diagnosis ranged from 18 to 96 years, with a median age of 62 years. While all skin

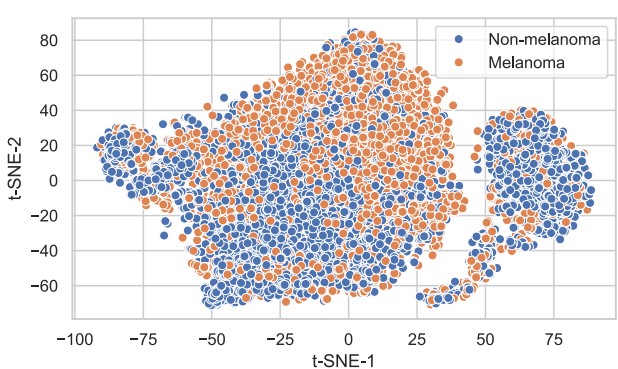

**Fig. 1 | Feature distribution of our dataset.** The feature distribution of our dataset ($n = 11460$) using $t$-distributed Stochastic Neighbor Embedding (2 components, 1000 iterations). The features are colored according to their type (melanoma or non-melanoma).

**Table 1 | Patient characteristics of our dataset**

| Patient characteristic | $n = 1716$ | % |
|---|---|---|
| Age [mean (std, IQR)] | 60.9 | (18.3, 28) |
| *Sex* | | |
| Male | 985 | 57.4 |
| Female | 731 | 42.6 |
| *Fitzpatrick skin type* | | |
| I | 138 | 8.04 |
| II | 1011 | 58.9 |
| III | 477 | 27.8 |
| IV | 37 | 2.16 |
| V | 3 | 0.175 |
| VI | 3 | 0.175 |
| Unknown | 47 | 2.74 |
| *Personal history of melanoma* | | |
| No | 1320 | 76.9 |
| Yes | 335 | 19.5 |
| Unknown | 61 | 3.55 |
| *Family history of melanoma* | | |
| No | 1495 | 87.1 |
| Yes | 123 | 7.17 |
| Unknown | 98 | 5.71 |
| *Number of enrolled lesions* | | |
| 1 | 1522 | 88.7 |
| 2 | 194 | 11.3 |

types are present in the study population, Fitzpatrick skin types II and III are most prevalent, whereas types V and VI are less frequent.

## Lesion characteristics

The dataset consists of 750 melanomas (including rare subtypes such as spitzoid or desmoplastic melanomas), 885 nevi, and 275 other diagnoses (for detailed lesion characteristics, see Supplementary Data 1). Additional information about the diagnosis, such as the exact subtype (see Table 4) and the self-assessed dermatologists' confidence in their diagnosis (on a scale from 1—low confidence to 5—high confidence), were collected. Furthermore, the lesion size and location were recorded. The lesions were collected from eight different university hospitals (i.e., the data source) with four distinct technical setups (i.e., hardware configurations, a certain combination of camera and dermatoscope that were used to capture the images, see Supplementary Methods), thus ensuring domain diversity.

## ADAE outperforms dermatologists in balanced accuracy and sensitivity

To assess the performance of AI algorithms in diagnosing suspected melanoma lesions in prospectively acquired data, we compared the prediction quality of ADAE to that of dermatologists recorded within physical patient examinations, who also had access to patient and lesion-specific metadata. This assessment is based primarily on balanced accuracy, and secondarily on sensitivity and specificity. $P$-values smaller than 0.05 are considered statistically significant and are determined using a pairwise two-sided Wilcoxon signed-rank test. The algorithm was enhanced with real test-time augmentation (R-TTA, introduced by Hekler et al. as MV-Real)[24] by providing multiple images per lesion at test-time (see also Supplementary Figs. 6 and 7). That is, each image was initially classified individually as either melanoma or non-melanoma (including basal and spinal cell carcinoma, dermatofibroma, keratosis, nevus, vascular lesions), and ultimately aggregated to produce one final prediction for the respective lesion. The performance was measured using histopathological labels diagnosed by experienced pathologists as ground truth.

Overall, at the predetermined 85% sensitivity threshold (see Methods/model training and testing), ADAE showed higher balanced accuracy than dermatologists originally diagnosing the lesions (ADAE: 0.798, 95% confidence interval (CI) 0.779–0.814 vs. dermatologists: 0.781, 95% CI 0.760–0.802; $p = 4.0e{-}145$) with significantly higher sensitivity (0.922, 95% CI 0.900–0.942 vs. 0.734, 95% CI 0.701–0.770; $p = 3.3e{-}165$), but significantly lower specificity (0.673, 95% CI 0.641–0.702 vs. 0.828, 95% CI 0.804–0.852; $p = 3.3e{-}165$). For the test set results, see Tables 2 and 3, and for differences stratified by domains, see Supplementary Figs. 1–4. In total, ADAE detected 602 of 653 melanomas (0.922 sensitivity), while dermatologists detected 479

of 653 (0.734 sensitivity), respectively (for the individual melanoma subtype results, see Table 4 and for a visualization of the differences, see Supplementary Fig. 3). Thereby, a total of 623 of 653 melanomas (0.954 sensitivity) were detected by either the AI, the dermatologist or both. Concurrently, ADAE classified 618 of 918 non-melanoma correctly (0.673 specificity), while dermatologists classified 760 of 918 non-melanoma (0.828 specificity) correctly. A total of 833 of 918 non-melanoma (0.907 specificity) were classified correctly by either the AI, the dermatologist, or both. Thus, the combination could exceed the individual outcomes of both AI and dermatologists, wherein solitary ADAE exhibited a higher detection, but also a higher false positive rate than dermatologists. Hence, a synergistic approach could benefit patients more than relying on man or machine alone.

## Subset analyses

Additionally, multiple subset analyses were performed to identify disparities in diagnostic performance, revealing substantial differences within certain subsets.

## Lesion subtypes

The algorithm showed significantly higher sensitivity on all melanoma subtypes except for nodular melanoma than dermatologists ($p < 0.001$ for all comparisons, see Table 4). It also showed significantly higher specificity for basal cell carcinoma, blue nevus, and actinic keratosis but significantly lower specificity for dysplastic/clark nevus, benign keratosis, and acral nevus ($p < 0.001$ for all comparisons, see Table 5).

## Data source

Moreover, the algorithm achieved a better-balanced accuracy than dermatologists on data derived from five out of seven hospitals (i.e., the data of the test set, excluding the validation data from hospital 8, see Table 3), but performed significantly worse on data from hospital 1 (0.772, 95% CI 0.742–0.803 vs. 0.783, 95% CI 0.751–0.817; $p = 6.7e{-}52$), and hospital 3

**Table 2 | Diagnostic performance of dermatologists and ADAE for biological subsets of our test set (i.e., validation samples are not included here)**

| Characteristic | Total lesions (n) | Total melanomas (n) | Dermatologists | | | ADAE | | | |
|---|---|---|---|---|---|---|---|---|---|
| | | | Bal. acc. | Sens. | Spec. | AUROC | Bal. acc. | Sens. | Spec. |
| Overall | 1571 | 653 | 0.781 | 0.734 | **0.828** | 0.916 | **0.798** | **0.922** | 0.673 |
| *Age (in years)* | | | | | | | | | |
| <35 | 177 | 14 | 0.767 | 0.571 | **0.963** | 0.974 | **0.890** | **1.000** | 0.779 |
| 35–54 | 372 | 109 | **0.821** | 0.780 | **0.863** | 0.931 | 0.818 | **0.945** | 0.692 |
| 55–74 | 571 | 287 | 0.776 | 0.739 | **0.813** | 0.913 | 0.793 | **0.913** | 0.673 |
| >74 | 451 | 243 | 0.707 | 0.716 | **0.697** | 0.879 | 0.743 | **0.918** | 0.567 |
| *Sex* | | | | | | | | | |
| Male | 896 | 399 | 0.760 | 0.734 | **0.785** | 0.909 | 0.783 | **0.920** | 0.646 |
| Female | 675 | 254 | 0.806 | 0.732 | **0.879** | 0.923 | 0.815 | **0.925** | 0.705 |
| *Location (grouped)* | | | | | | | | | |
| Posterior torso | 469 | 196 | **0.791** | 0.765 | **0.817** | 0.920 | 0.788 | **0.923** | 0.652 |
| Anterior torso | 294 | 80 | 0.804 | 0.725 | **0.883** | 0.951 | **0.833** | **0.938** | 0.729 |
| Lower extremity | 293 | 116 | **0.856** | 0.853 | **0.859** | 0.935 | 0.817 | **0.957** | 0.678 |
| Head/neck | 278 | 137 | 0.660 | 0.562 | **0.759** | 0.881 | **0.775** | **0.883** | 0.667 |
| Upper extremity | 191 | 111 | 0.778 | 0.757 | **0.800** | 0.893 | **0.784** | **0.919** | 0.650 |
| Palms/soles | 29 | 11 | **0.798** | 0.818 | **0.778** | 0.798 | 0.649 | **0.909** | 0.389 |
| Oral/genital | 12 | 1 | **0.864** | 1.000 | **0.727** | 0.818 | 0.818 | 1.000 | 0.636 |
| Unknown | 5 | 1 | 0.875 | 1.000 | 0.750 | 1.000 | **1.000** | 1.000 | **1.000** |
| *Diameter (in mm)* | | | | | | | | | |
| ≦3.00 | 120 | 14 | **0.784** | 0.643 | **0.925** | 0.806 | 0.772 | **0.714** | 0.830 |
| 3.01–6.00 | 387 | 62 | 0.660 | 0.419 | **0.902** | 0.797 | **0.715** | **0.726** | 0.705 |
| 6.01–9.00 | 274 | 89 | 0.695 | 0.584 | **0.805** | 0.895 | **0.796** | **0.921** | 0.670 |
| 9.01–12.00 | 313 | 155 | 0.782 | 0.735 | **0.829** | 0.924 | **0.803** | **0.929** | 0.677 |
| 12.01–15.00 | 151 | 94 | 0.713 | 0.777 | **0.649** | 0.918 | **0.765** | **0.968** | 0.561 |
| >15.00 | 326 | 239 | **0.728** | 0.858 | **0.598** | 0.890 | 0.700 | **0.962** | 0.437 |
| *Fitzpatrick skin type* | | | | | | | | | |
| I | 149 | 77 | **0.797** | 0.857 | **0.736** | 0.865 | 0.760 | **0.922** | 0.597 |
| II | 797 | 390 | 0.778 | 0.708 | **0.848** | 0.912 | **0.796** | **0.918** | 0.673 |
| III | 531 | 160 | 0.774 | 0.731 | **0.817** | 0.938 | **0.821** | **0.931** | 0.712 |
| IV | 39 | 9 | **0.839** | 0.778 | **0.900** | 0.985 | 0.783 | **1.000** | 0.567 |
| V | 3 | 1 | **1.000** | 1.000 | **1.000** | 1.000 | 0.750 | 1.000 | 0.500 |
| VI | 2 | 0 | **0.500** | – | **0.500** | – | 0.000 | – | 0.000 |
| Unknown | 50 | 16 | **0.801** | 0.750 | **0.853** | 0.877 | 0.717 | **0.875** | 0.559 |
| *Pigmented lesion* | | | | | | | | | |
| Yes | 1817 | | 0.780 | 0.731 | **0.830** | 0.920 | **0.801** | **0.930** | 0.671 |
| No | 51 | | **0.841** | **0.875** | **0.806** | 0.673 | 0.542 | 0.375 | 0.710 |
| Unknown | 42 | | 0.774 | 0.786 | **0.762** | 0.878 | **0.786** | **0.857** | 0.714 |

Best performance metrics are each highlighted with bold text. *Bal. acc.* balanced accuracy, *Sens.* sensitivity, *Spec.* specificity, *AUROC* area under the receiver operating characteristic curve.

(0.615, 95% CI 0.521–0.716 vs. 0.758, 95% CI 0.659–0.860; $p = 7.4e{-}165$). The dermatologists' sensitivity was worse for all data except those from hospital 3 (0.897, 95% CI 0.806–0.976 vs. 0.923, 95% CI 0.823–1.000; $p = 6.7e{-}23$), while their specificity was higher without exception. Although consistent with the overall test set results, this observation highlights Hospital 3 as an outlier.

**Patient age and lesion location**

Furthermore, ADAE achieved significantly higher balanced accuracy in patients younger than 35 years (0.890, 95% CI 0.859–0.920 vs. 0.767, 95% CI 0.636–0.897; $p = 1.9e{-}161$), and for lesions on the head or neck (0.775, 95% CI 0.726–0.822 vs. 0.660, 95% CI 0.603–0.714; $p = 3.3e{-}165$) but performed significantly worse for lesions on the palms or soles (0.649, 95% CI 0.508–0.774 vs. 0.798, 95% CI 0.642–0.944; $p = 2.0e{-}158$) compared to dermatologists.

**Classification complexity**

The dermatologists indicated the level of confidence they had in their own diagnosis on a scale from 1 (low confidence) to 5 (high confidence), signifying the perceived classification complexity of the lesion. Based on these data, the algorithm demonstrated significantly higher balanced accuracy than the dermatologists on lesions that were assigned lower confidence scores (confidence score 1: 0.754, 95% CI 0.608–0.895 vs. 0.508, 95% CI 0.357–0.688; $p = 3.3e{-}164$; confidence score 2: 0.761, 95% CI 0.689–0.829 vs. 0.588, 95% CI 0.491–0.680; $p = 3.3e{-}165$; confidence score 3: 0.767, 95% CI 0.729–0.804 vs. 0.662, 95% CI 0.615–0.709; $p = 3.3e{-}165$;

**Table 3 | Diagnostic performance of dermatologists and ADAE for other subsets of our test set (i.e., validation samples are not included here)**

| Characteristic | Total lesions (*n*) | Total melanomas (*n*) | Dermatologists | | | ADAE | | | |
|---|---|---|---|---|---|---|---|---|---|
| | | | Bal. acc. | Sens. | Spec. | AUROC | Bal. acc. | Sens. | Spec. |
| Overall | 1571 | 653 | 0.781 | 0.734 | **0.828** | 0.916 | **0.798** | **0.922** | 0.673 |
| *Hospital* | | | | | | | | | |
| Hospital 1 | 567 | 245 | **0.783** | 0.669 | **0.898** | 0.901 | 0.772 | **0.914** | 0.630 |
| Hospital 2 | 265 | 123 | 0.764 | 0.709 | **0.819** | 0.922 | **0.810** | **0.891** | 0.729 |
| Hospital 3 | 66 | 39 | 0.758 | **0.923** | **0.593** | 0.775 | 0.615 | 0.897 | 0.333 |
| Hospital 4 | 215 | 93 | 0.765 | 0.748 | **0.783** | 0.937 | **0.828** | **0.959** | 0.696 |
| Hospital 5 | 106 | 31 | 0.800 | 0.774 | **0.827** | 0.932 | **0.852** | **0.903** | 0.800 |
| Hospital 6 | 176 | 67 | 0.739 | 0.716 | **0.761** | 0.890 | **0.771** | **0.881** | 0.661 |
| Hospital 7 | 176 | 55 | 0.806 | 0.817 | **0.795** | 0.934 | **0.822** | **0.957** | 0.687 |
| *Technical domain* | | | | | | | | | |
| Setup 1 | 898 | 339 | 0.779 | 0.705 | **0.853** | 0.903 | **0.781** | **0.909** | 0.653 |
| Setup 2 | 321 | 154 | 0.778 | 0.753 | **0.802** | 0.943 | **0.845** | **0.948** | 0.743 |
| Setup 3 | 352 | 160 | 0.776 | 0.775 | **0.776** | 0.912 | **0.798** | **0.925** | 0.672 |
| *Dermatologists' confidence rating* | | | | | | | | | |
| 1 | 36 | 13 | 0.508 | 0.538 | 0.478 | 0.913 | **0.754** | **0.769** | **0.739** |
| 2 | 111 | 34 | 0.588 | 0.500 | **0.675** | 0.851 | **0.761** | **0.912** | 0.610 |
| 3 | 424 | 141 | 0.662 | 0.546 | **0.777** | 0.860 | **0.767** | **0.872** | 0.661 |
| 4 | 575 | 206 | 0.806 | 0.733 | **0.878** | 0.934 | **0.811** | **0.942** | 0.680 |
| 5 | 425 | 259 | **0.899** | 0.876 | **0.922** | 0.939 | 0.820 | **0.942** | 0.699 |
| *AI confidence rating* | | | | | | | | | |
| 1 | 240 | 67 | **0.624** | 0.433 | **0.815** | 0.531 | 0.526 | **0.821** | 0.231 |
| 2 | 371 | 107 | 0.653 | 0.514 | **0.792** | 0.735 | **0.679** | **0.832** | 0.527 |
| 3 | 309 | 140 | 0.794 | 0.736 | 0.852 | 0.904 | **0.889** | **0.914** | **0.864** |
| 4 | 419 | 202 | 0.813 | 0.787 | 0.839 | 0.948 | **0.943** | **0.955** | **0.931** |
| 5 | 232 | 137 | 0.928 | 0.971 | 0.884 | 0.990 | **0.979** | **1.000** | **0.958** |

Best performance metrics are each highlighted with bold text. *Bal. acc.* balanced accuracy, *Sens.* sensitivity, *Spec.* specificity, *AUROC* area under the receiver operating characteristic curve.

**Table 4 | Detection rate of melanoma, stratified by melanoma subtype**

| Melanoma subtype | Dermatologist | | | ADAE | | | Total | *p*-Value |
|---|---|---|---|---|---|---|---|---|
| | *n* | Sens. | 95% CI | n | Sens. | 95% CI | n | |
| Superficial spreading | 228 | 0.820 | 0.773–0.863 | **266** | **0.957** | 0.932–0.978 | 278 | 3.0e−165 |
| Other | 140 | 0.593 | – | 206 | 0.873 | – | 236 | – |
| Nodular | **56** | **0.982** | 0.947–1.000 | 55 | 0.965 | 0.912–1.000 | 57 | 9.8e−57 |
| Lentigo maligna | 27 | 0.600 | 0.466–0.733 | **41** | **0.911** | 0.822–0.978 | 45 | 2.5e−165 |
| Acral lentiginous | 16 | 0.842 | 0.684–1.000 | **17** | **0.895** | 0.737–1.000 | 19 | 1.4e−60 |
| Combined forms of melanoma | 12 | 0.667 | 0.444–0.889 | **17** | **0.944** | 0.833–1.000 | 18 | 6.4e−163 |
| Total | 479 | 0.734 | | **602** | **0.922** | | 653 | |

Best performance metrics are highlighted with bold text.

confidence score 4: 0.811, 95% CI 0.782–0.839 vs. 0.805, 95 CI 0.775–0.838; *p* = 5.8e−18) but lower balanced accuracy for lesions diagnosed with a confidence score of 5 (0.820, 95% CI 0.779–0.858 vs. 0.899, 95% CI 0.871–0.925; *p* = 3.3e−165). Hence, the AI algorithm seems less susceptible to the diagnostic difficulty. This also indicates that dermatologists assess their own prediction somewhat accurately, and especially when unsure could benefit greatly from such an AI prediction.

Likewise, we binned the confidence of the AI algorithm into a comparable 1 (low confidence) to 5 (high confidence) scale. AI and dermatologists' confidence ratings have low correlation (see Supplementary Fig. 8). Based on this stratification, the algorithm demonstrated significantly higher balanced accuracy than dermatologists on lesions with AI confidence scores of 2 or higher (AI confidence score 2: 0.679, 95% CI 0.634–0.727 vs. 0.653, 95% CI 0.603–0.705; *p* = 1.4e−96; AI confidence score 3: 0.889, 95% CI 0.851–0.921 vs. 0.794, 95% CI 0.748–0.835; *p* = 3.3e−165; AI confidence score 4: 0.943, 95% CI 0.920–0.962 vs. 0.813, 95% CI 0.776–0.851; *p* = 3.3e−165; AI confidence score 5: 0.979, 95% CI 0.956–0.995 vs. 0.928, 95% CI 0.891–0.962; *p* = 3.3e−165) but lower balanced accuracy for lesions with an AI confidence score of 1 (0.526, 95% CI 0.470–0.579 vs. 0.624, 95% CI 0.558–0.691; *p* = 9.0e−165). When taking the confidence of the AI algorithm into account, there lies potential in more accurate and trustworthy predictions.

**Table 5 | Detection rate of non-melanoma, stratified by non-melanoma subtype**

| Melanoma subtype | Dermatologist | | | ADAE | | | Total | p-Value |
|---|---|---|---|---|---|---|---|---|
| | **n** | **Spec.** | **95% CI** | **n** | **Spec.** | **95% CI** | **n** | |
| Others | 327 | 0.841 | – | 278 | 0.715 | – | 389 | – |
| Dysplastic/clark nevus | **295** | **0.905** | 0.871–0.936 | 206 | 0.631 | 0.583–0.684 | 326 | 3.1e−165 |
| Benign keratosis | **51** | **0.680** | 0.573–0.774 | 46 | 0.613 | 0.506–0.720 | 75 | 1.9e−111 |
| Basal cell carcinoma | 25 | 0.510 | 0.367–0.653 | **30** | **0.612** | 0.469–0.735 | 49 | 1.7e−145 |
| Blue nevus | 26 | 0.788 | 0.636–0.909 | **29** | **0.879** | 0.758–0.970 | 33 | 1.1e−127 |
| Acral nevus | **25** | **0.962** | 0.885–1.000 | 16 | 0.615 | 0.423–0.808 | 26 | 1.2e−67 |
| Actinic keratosis | 11 | 0.550 | 0.349–0.750 | **13** | **0.650** | 0.450–0.850 | 20 | 1.2e−67 |
| Total | **760** | **0.828** | | 618 | 0.673 | | 918 | |

Best performance metrics are highlighted with bold text.

### ADAE generalizes robustly on different subsets

The predictive quality of AI algorithms may depend on certain features of the test set, such as data source, patient age, lesion size, or location. To analyze whether the algorithm in this study demonstrates robust generalizability, we evaluated the association between the predictions and specific stratified domains included in the heterogeneous dataset used for classifier testing (see Tables 2 and 3 for test set characteristics). Therefore, the Breslow–Day test for homogeneity of the odds ratio was used. For inter-AI comparisons, the area under the receiver operating characteristic curve (AUROC), in combination with DeLong's method, is used as a more objective alternative than accuracy since it is independent of a threshold setting. $p$-Values smaller than 0.05 are considered statistically significant.

### ADAE prediction generalizability

Overall, there was no significant association between the ADAE prediction performance and the patient age ($p = 0.104$), skin type ($p = 0.587$; excluding unknown skin types, and skin types V + VI due to low sample size (<25)), or lesion location domain ($p = 0.233$; excluding lesions in unknown locations, and oral/genital lesions due to low sample size (<25)), the technical domain (i.e., camera setup; $p = 0.068$) or the dermatologists' diagnostic confidence score, although borderline significant ($p = 0.050$). However, a significant association between lesion diameter ($p = 0.009$) and data source (i.e., the originating hospital; $p = 0.027$) was identified, indicating that performance correlates with these features. Specifically, the algorithm performed significantly worse on data from hospital 3, as indicated by the lower AUROC score (hospital 3: 0.775, 95% CI 0.660–0.877 vs. other hospitals combined: 0.921, 95% CI 0.906–0.934; $p = 0.013$). Without the outlier dataset (hospital 3), there was no significant association between the predictive performance and data source ($p = 0.416$). Furthermore, the AUROC was significantly higher for lesions with diameters greater than 6 mm (≤6 mm: 0.802, 95% CI 0.747–0.857 vs. >6 mm: 0.917, 95% CI 0.901–0.932; $p = 7.2e−5$), and significantly higher for pigmented lesions (pigmented: 0.920, 95% CI 0.905–0.934, vs. non-pigmented: 0.673, 95% CI 0.469–0.859; $p = 0.019$). These findings suggest that the algorithm has robust generalization capabilities on most domains, while it is influenced by lesion diameter, pigmentation and data source. However, it is worth noting that some meta information, such as patient age and lesion location, is used by ADAE as input.

While not all differences exhibited by ADAE within each subset were significant, the disparities between the upper and lower bound within some subsets were significant. Specifically, for patient age, ADAE achieved a higher AUROC score for the youngest patients than for the oldest ones (<35: 0.974, 95% CI 0.942–0.997 vs. >74: 0.879, 95% CI 0.847–0.909; $p = 1.1e−5$). Similarly, for the Fitzpatrick skin type, the lightest skin type was associated with lower AUROCs than the darkest one (type I: 0.865, 95% 0.802–0.922 vs. type IV: 0.985, 95% CI 0.943–1.000; $p = 4.0e−4$). Thus, even though the algorithm generalizes robustly, certain trends are still evident, signifying

partial dependencies with respect to some groups of lesions or patients, which can still affect diagnostic performance.

### Dermatologist prediction generalizability

Among dermatologists, on the other hand, there were no significant associations between prediction performance and skin type (excluding unknown skin types and V + VI due to low sample size (<25); $p = 0.750$), lesion diameter ($p = 0.164$), pigmentation ($p = 0.781$), technical domain ($p = 0.862$), or data source ($p = 0.527$). There were, however, significant associations of the prediction performance with patient age ($p = 2.9e−4$), lesion location (excluding unknown locations, and oral/genital locations due to low sample size (<25); $p = 7.5e−6$), and the dermatologist diagnostic confidence ($p < 1.0e−6$). These findings indicate that the prediction quality of the dermatologists depends on these features. Specifically, the dermatologists' specificity decreased with increasing patient age (<35 years: 0.963, 95% CI 0.932–0.988 vs. 35–54 years: 0.863, 95% CI 0.819–0.902; $p = 3.3e−165$; vs. 55–74 years: 0.813, 95% CI 0.766–0.859; $p = 1.3e−157$; vs. >74 years: 0.697, 95% CI 0.632–0.757; $p = 3.3e-165$). Relatedly, sensitivity was significantly lower for patients younger than 35 years (<35 years: 0.571, 95% CI 0.308-0.833 vs. all other age groups combined: 0.737, 95% CI 0.703–0.771; $p = 3.7e−137$) but followed similar trends as the specificity for age groups older than 35 years (35–54 years: 0.780, 95% CI 0.692–0.850 vs. 55–74 years: 0.739, 95% CI 0.688–0.788; $p = 1.0e−96$; vs. >74 years: 0.716, 95% CI 0.659–0.773; $p = 8.8e−53$). Additionally, the dermatologists' balanced accuracy was significantly lower for lesions on the head or neck (head/neck: 0.660, 95% CI 0.603–0.714 vs. all other locations combined: 0.810, 95% CI 0.788–0.832; $p = 3.3e−165$) but was higher for lesions that received higher dermatologists' confidence scores (confidence score 1: 0.508, 95% CI 0.357–0.688 vs. confidence score 2: 0.588, 95% CI 0.491–0.680; $p = 1.3e−86$; vs. confidence score 3: 0.662, 95% CI 0.615–0.709; $p = 1.2e−149$; vs. confidence score 4: 0.806, 95% CI 0.775–0.838; $p = 3.3e−165$; vs. confidence score 5: 0.899, 95% CI 0.871–0.925; $p = 3.3e−165$).

### Discussion

In this multicenter study with prospectively collected samples, we evaluated the diagnostic performance of ADAE in differentiating between melanoma and non-melanoma skin lesions and compared it to dermatologists' diagnostic accuracy as recorded during real-life patient examinations. One of the main strengths of the study, in addition to the prospective data collection, is its comprehensive test set, which includes a wide range of melanoma subtypes and lesion locations encountered in routine care. Altogether, ADAE performed better than dermatologists in terms of balanced accuracy and sensitivity but achieved a lower specificity. Moreover, the algorithm generalized robustly to domains such as patient age and skin type, lesion location, and camera setup, but its performance was affected by lesion diameter. The dermatologists' diagnostic accuracy, in contrast, correlated significantly with patient age and lesion location.

To evaluate AI algorithms for melanoma detection, we measured the performance of ADAE against dermatologists using a prospectively collected, heterogeneous dataset. ADAE performed better in terms of balanced accuracy (ADAE: 0.798 vs. dermatologists: 0.781), and achieved a higher sensitivity (0.922 vs. 0.734) at the cost of a lower specificity (0.673 vs. 0.828). Additionally, our findings suggest that AI algorithms may be better suited than dermatologists for diagnosing skin lesions of younger patients or patients with lesions on the head or neck, as indicated by the balanced accuracy in these tasks (<35 years: 0.890 vs. 0.767 and head/neck: 0.775 vs. 0.660, respectively). In contrast, dermatologists were significantly better at diagnosing lesions on acral skin, i.e., on the palms or soles (0.649 vs. 0.798). Interestingly, in our study, the algorithm exhibited significantly higher diagnostic accuracy in cases where dermatologists tended to be unsure, and vice-versa, thus highlighting the potential synergies between AI and human experts. Our findings are in line with previous studies[7,10,11,13,31–33] that demonstrated the potential advantages arising from the cooperation of dermatologists with AI. Our study addresses the limitations of previous studies, specifically by our multicenter design encompassing a substantial number of dermatologists, a larger cohort of lesions, and rare melanoma subtypes. Marchetti et al.[13] previously analyzed ADAE in terms of classification performance and its impact on dermatologists' decisions, but are limited by a small test set. We compare our results to underscore their external validity. Our overall AUROC was slightly higher than Marchetti et al. reported (without R-TTA: 0.898, 95% CI 0.884–0.913 vs. 0.858). The subset analyses were also largely similar between the studies: the performance of ADAE was worse for older patients and those with type I skin in both studies. Specificity was lower for the larger lesions, despite the increase in AUROC. However, unlike in our study, Marchetti et al. reported a lower specificity of the algorithm for head/neck area lesions. These findings suggest that a collaborative[13,31] rather than a comparative approach[7,10,11,32] may ultimately lead to an improvement in patient care, achieving an increased detection rate while reducing the number of unnecessary excisions as compared to relying solely on either dermatologists or AI alone.

When we investigated the effect of the different variables on diagnostic performance, we found differences in those effects between the AI algorithm and dermatologists. Specifically, the performance of the AI was affected by the lesion diameter and pigmentation; it performed worse for lesions with diameters smaller than 6 mm, and for lesions without pigmentation, while dermatologists discriminated lesions of all sizes and pigmentation states relatively consistently. Concurrently, the dermatologists' decisions, but not those of ADAE, were influenced by patient age and lesion location. This further underscores the advantages of a holistic approach, as the diagnostic strengths of the AI and dermatologists may compensate for each other's shortcomings in their generalization abilities.

While the algorithm was largely unaffected by the data source (i.e., the hospital), it performed significantly worse on data from one particular hospital, hospital 3. While the sensitivity is comparable to other hospitals (hospital 3: 0.897 vs. other hospitals: 0.923), the specificity is significantly worse (0.333 vs. 0.684). One contributing factor is the presence of relatively larger lesions (mean of 14.3 mm vs. 12.3 mm). Additionally, the proportion of non-pigmented skin lesions is higher (10.6% vs. 2.13%). Furthermore, the population of this hospital comprised older individuals when compared to the other hospitals (age at diagnosis: 27–96, median of 65.5 years vs. 18–95, median of 63 years) while exhibiting a slightly different distribution of skin types from the overall study population (i.e., a preponderance of type I and II skin types). We did find that ADAE performed worse for non-pigmented skin lesions, older patients and those with lighter skin types, and has lower specificity the larger the lesion, which may explain the deviant performance.

While our study comprises multiple centers, they are all located in Germany. Thus, our findings might not translate to other ethnicities or skin types (especially types V and VI, which are underrepresented in our study). Furthermore, we are limited to a binary classification (melanoma vs. non-melanoma), which does not fully model the complexity of clinical reality which involves lesion classification into multiple classes. Finally, our comparison of AI and dermatologists was performed by comparing diagnostic

accuracy and generalization but does not include other metrics and aspects, such as the for ensembles typically problematic computing costs and explainability, nor does it investigate a prospective impact on dermatologists' management decisions. Especially explainability is a feature of AI that is both required by EU standards for transparency in AI[34,35] and demanded by physicians and patients alike[36–39]. In a recent retrospective study, a dermatologist-like explainable AI-enhanced trust and confidence in diagnosing melanoma among 116 participating clinicians, promoting its future use in care[33]. Future research could build upon this work by evaluating different AI architectures, such as model soups[40,41], that use the average weights of multiple models to improve performance and address the ensemble drawbacks, namely performance costs as well as explainability aspects.

In conclusion, ADAE showed better performance than dermatologists in terms of balanced accuracy and sensitivity, but worse specificity. It generalizes robustly on most domains of a heterogeneous, prospectively collected test set. Thus, it could be particularly useful in medical settings, where there often is a large discrepancy between hospitals due to technical differences related to imaging and sometimes patient populations. Ultimately, AI algorithms can support physicians in their diagnoses to identify melanomas more accurately, especially for difficult cases in which human dermatologists are unsure of their diagnoses. Future research should address the shortcomings of the algorithm, such as lack of explainability and low specificity, both particularly problematic in facilitating the clinical use of AI.

## Data availability
The SIIM-ISIC 2020 data is publicly available at https://challenge2020.isic-archive.com. Our dataset is available from the corresponding author on reasonable request. The numerical data underlying Fig. 1 (source data) can be found in Supplementary Data 2.

## Code availability
The pretrained ADAE algorithm is publicly available via GitHub https://github.com/ISIC-Research/ADAE[23].

## Abbreviations

| | |
|---|---|
| AI | artificial intelligence |
| ADAE | All Data Are Ext |
| SIIM | Society for Imaging Informatics in Medicine |
| ISIC | International Skin Image Collaboration |
| R-TTA | real test-time augmentation |
| CI | confidence interval |
| AUROC | area under the receiver operating characteristic curve |
| CNN | convolutional neural network |

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

## Acknowledgements

This study was funded by the Federal Ministry of Health, Berlin, Germany (grant: Skin Classification Project 2; grant holder: Titus J. Brinker, German Cancer Research Center, Heidelberg, Germany). The sponsor had no role in the design and conduct of the study; collection, management, analysis and interpretation of the data; preparation, review, or approval of the paper; and decision to submit the paper for publication.

## Author contributions

Dr Brinker had full access to all the data in the study and took responsibility for the integrity of the data and the accuracy of the data analysis. These authors contributed equally: Lukas Heinlein, Achim Hekler, and Roman Maron. *Concept and design:* Hekler, Maron, Brinker, Heinlein, Haggenmüller. *Acquisition, analysis, or interpretation of data:* All authors. *Drafting of the paper:* Hekler, Maron, Heinlein. *Critical revision of the manuscript for important intellectual content:* All authors. *Statistical analysis:* Hekler, Maron, Heinlein, Haggenmüller, Wies. *Obtained funding:* Brinker. *Administrative, technical, or material support:* Hekler, Maron, Haggenmüller, Brinker. *Supervision:* Hekler, Haggenmüller, Brinker.

## Funding

## Competing interests

Jochen S. Utikal is on the advisory board or has received honoraria and travel support from Amgen, Bristol Myers Squibb, GSK, Immunocore, LeoPharma,

Merck Sharp and Dohme, Novartis, Pierre Fabre, Roche, and Sanofi outside the submitted work. Friedegund Meier has received travel support and/or speaker's fees and/or advisor's honoraria by Novartis, Roche, BMS, MSD and Pierre Fabre and research funding from Novartis and Roche. Sarah Hobelsberger reports clinical trial support from Almirall and speaker's honoraria from Almirall, UCB, and AbbVie and has received travel support from the following companies: UCB, Janssen Cilag, Almirall, Novartis, Lilly, LEO Pharma and AbbVie outside the submitted work. Sebastian Haferkamp reports advisory roles for or has received honoraria from Pierre Fabre Pharmaceuticals, Novartis, Roche, BMS, Amgen, and MSD outside the submitted work. Konstantin Drexler has received honoraria from Pierre Fabre Pharmaceuticals and Novartis. Axel Hauschild reports clinical trial support, speaker's honoraria, or consultancy fees from the following companies: Agenus, Amgen, BMS, Dermagnostix, Highlight Therapeutics, Immunocore, Incyte, IO Biotech, Merck Pfizer, MSD, NercaCare, Novartis, Philogen, Pierre Fabre, Regeneron, Roche, Sanofi-Genzyme, Seagen, Sun Pharma and Xenthera outside the submitted work. Lars E. French is on the advisory board or has received consulting/speaker honoraria from Galderma, Janssen, Leo Pharma, Eli Lilly, Almirall, Union Therapeutics, Regeneron, Novartis, Amgen, AbbVie, UCB, Biotest, and InflaRx. Max Schlaak reports advisory roles for Bristol-Myers Squibb, Novartis, MSD, Roche, Pierre Fabre, Kyowa Kirin, Immunocore, and Sanofi-Genzyme. Wiebke Sondermann reports grants, speaker's honoraria, or consultancy fees from medi GmbH Bayreuth, AbbVie, Almirall, Amgen, Bristol-Myers Squibb, Celgene, GSK, Janssen, LEO Pharma, Lilly, MSD, Novartis, Pfizer, Roche, Sanofi Genzyme and UCB outside the submitted work. Bastian Schilling reports advisory roles for or has received honoraria from Sanofi, Pierre Fabre Pharmaceuticals, SUN Pharma, and BMS, and research funding from Novartis, all outside the submitted work. Matthias Goebeler has received speaker's honoraria and/or has served as a consultant and/or member of advisory boards for Almirall, Argenx, Biotest, Eli Lilly, Janssen Cilag, Leo Pharma, Novartis, and UCB outside the submitted work. Michael Erdmann declares honoraria and travel support from Bristol-Myers Squibb, Immunocore Novartis, Pierre Farbe, and Sanofi outside the submitted work. Jakob N. Kather reports consulting services for Owkin, France, Panakeia, UK, and DoMore Diagnostics, Norway, and has received honoraria for lectures by MSD, Eisai, and Fresenius. Titus J. Brinker reports owning a company that develops mobile apps (Smart Health Heidelberg GmbH, Handschuhsheimer Landstr. 9/1, 69120 Heidelberg). The remaining authors declare no competing interests.

## Additional information

Lukas Heinlein [1,16], Roman C. Maron [1,16], Achim Hekler [1,16], Sarah Haggenmüller[1], Christoph Wies [1,2], Jochen S. Utikal [3,4,5], Friedegund Meier [6,7], Sarah Hobelsberger[6], Frank F. Gellrich[6], Mildred Sergon[6], Axel Hauschild[8], Lars E. French [9,10], Lucie Heinzerling[9,11], Justin G. Schlager[9], Kamran Ghoreschi [12], Max Schlaak [12], Franz J. Hilke[12], Gabriela Poch[12], Sören Korsing[12], Carola Berking [11], Markus V. Heppt [11], Michael Erdmann [11], Sebastian Haferkamp[13], Konstantin Drexler[13], Dirk Schadendorf [14], Wiebke Sondermann [14], Matthias Goebeler [15], Bastian Schilling [15], Eva Krieghoff-Henning[1] & Titus J. Brinker [1] ✉

[1]Digital Biomarkers for Oncology Group, German Cancer Research Center (DKFZ), Heidelberg, Germany. [2]Medical Faculty, University Heidelberg, Heidelberg, Germany. [3]Department of Dermatology, Venereology and Allergology, University Medical Center Mannheim, Ruprecht-Karl University of Heidelberg, Mannheim, Germany. [4]Skin Cancer Unit, German Cancer Research Center (DKFZ), Heidelberg, Germany. [5]DKFZ Hector Cancer Institute at the University Medical Center Mannheim, Mannheim, Germany. [6]Department of Dermatology, Faculty of Medicine and University Hospital Carl Gustav Carus, Technische Universität Dresden, Dresden, Germany. [7]Skin Cancer Center at the University Cancer Centre Dresden and National Center for Tumor Diseases, Dresden, Germany. [8]Department of Dermatology, University Hospital (UKSH), Kiel, Germany. [9]Department of Dermatology and Allergy, University Hospital, LMU Munich Munich, Germany. [10]Dr. Phillip Frost Department of Dermatology and Cutaneous Surgery, University of Miami, Miller School of Medicine, Miami, FL, USA. [11]Department of Dermatology, University Hospital Erlangen, Comprehensive Cancer Center Erlangen—European Metropolitan Region Nürnberg, CCC Alliance WERA, Erlangen, Germany. [12]Department of Dermatology, Venereology and Allergology, Charité—Universitätsmedizin Berlin, Corporate member of Freie Universität Berlin and Humboldt-Universität zu Berlin, Berlin, Germany. [13]Department of Dermatology, University Hospital Regensburg, Regensburg, Germany. [14]Department of Dermatology, Venereology and Allergology, University Hospital Essen, University Duisburg-Essen, Essen, Germany. [15]Department of Dermatology, Venereology and Allergology, University Hospital Würzburg and National Center for Tumor Diseases (NCT) WERA, Würzburg, Germany. [16]These authors contributed equally: Lukas Heinlein, Roman C. Maron, Achim Hekler. ✉e-mail: titus.brinker@dkfz.de

