## [Peer Review File · Communications Medicine]

Reviewers' comments:

Reviewer #1 (Remarks to the Author):

The authors provide an excellent contribution to the development of applications for artificial intelligence models in melanoma early-detection. The presented data constitute one of the most comprehensive analyses of the effectiveness of the “All Data are Ext” (ADAE) algorithm in diagnosing of melanoma, surpassing the limitations of the existing literature (limited samples, inclusion of limited number of dermatologists and lack of standardization).

Minor Remarks: no further details are available for the 275 "other" non-melanocytic lesions. It would be helpful to provide, if available, more information about their nature and the diagnostic accuracy of the ADAE algorithm in the differential diagnosis of individual types of lesions compared to melanoma, even if it is not within the main objective of the study.

Moreover, it would be also useful to provide more information and/or hypothesis on the confounding factors that led to significantly different results in specific sub-settings (i.e. hospital 3): could it be influenced by the smaller number of the lesions? How many rare variant of melanoma or non-pigmented lesions were included?

Reviewer #2 (Remarks to the Author):

The author conducted a prospective study to assess the effectiveness of an AI model in detecting melanoma, utilizing a homogeneous dataset that encompasses multiple melanoma subtypes sourced from various centres. The study focused on evaluating the model's generalization capabilities and benchmarking its performance against that of dermatologists. The experimental findings reveal that the AI model achieved a balanced accuracy superior to that of dermatologists, demonstrating its potential as a supportive tool for dermatologists, especially in diagnosing complex cases.

Detailed Comments:

Melanoma subtypes: the author train the Model on the public ISIC dataset with binary labels, how does it was applied to this multiple melanoma subtypes analysis? And Table 4 presents only the sensitivity, but it is not clear what the case that lesion was misclassified. To enhance clarity and

insight into the model's performance across different melanoma subtypes, it would be beneficial to include a confusion matrix in the subtype analysis.

While the test dataset incorporates a variety of rare melanoma classes, the number of images for some classes is notably low—for instance, certain classes are represented by merely three samples. This limited sample size could undermine the credibility of the results.

Classification complexity: the author evaluated the performance of AI against dermatologists by categorizing lesion complexity, which was determined based on the confidence levels of the dermatologists. However, the study lacks detailed information regarding the experience of the dermatologists involved and the methodology behind the human evaluation process. For a more comprehensive understanding, it would be beneficial to include details such as the years of experience of the dermatologist.

Additionally, it is unclear whether the dermatologist had access to meta-information (such as age, sex, family history) during their diagnosis, which would more closely mimic a real-world diagnostic setting. Incorporating these details would provide a clearer picture of the study's context and the basis for the comparison between AI and dermatologist performance.

The results indicate that the AI algorithm appears to be less affected by diagnostic challenges. However, redefining lesion difficulty based on the AI's confidence level and then comparing this to human assessments could provide valuable insights. Such an approach would be instrumental in developing more effective strategies for human-AI collaboration. Additionally, the author could enhance the study by presenting examples that showcase the diagnostic preferences between the AI algorithm and dermatologists in assessing lesions.

Generalization: The author asserts that the model exhibits robustness with respect to age, sex, and lesion location, yet it's important to note that the model was trained using these very variables as inputs. It may influence the conclusions drawn about the model's ability to generalize across these variables if it relies on them for evaluation. A more thorough analysis might be needed to ascertain the model's true generalizability.

The model's performance is significantly worse than that of dermatologist (~14% in BACC) on the data from hospital 3, however, the author did not provide sufficient analysis on this. If it is affected by age, then what is the detailed age distribution, how the error prediction distributed within the data? Moreover, the author did not provide any qualitative analysis or visualization analysis for the model. e.g., umap to illustrate the feature distribution for the heterogeneous data.

Response to Reviewers

We sincerely thank the reviewers for their diligent assessment of our work and for their insightful comments and recommendations. In light of their feedback, we have made substantial changes to the manuscript, which we believe have strengthened its overall quality and addressed the reviewers' suggestions.

Reviewer #1 (Remarks to the Author):

The authors provide an excellent contribution to the development of applications for artificial intelligence models in melanoma early-detection. The presented data constitute one of the most comprehensive analyses of the effectiveness of the "All Data are Ext" (ADAE) algorithm in diagnosing of melanoma, surpassing the limitations of the existing literature (limited samples, inclusion of limited number of dermatologists and lack of standardization).

Minor Remarks: no further details are available for the 275 "other" non-melanocytic lesions. It would be helpful to provide, if available, more information about their nature and the diagnostic accuracy of the ADAE algorithm in the differential diagnosis of individual types of lesions compared to melanoma, even if it is not within the main objective of the study.

Thank you very much for your thoughtful observation. We have incorporated the distribution of non-melanocytic lesions in Table 2 of the revised manuscript. Moreover, we have added information about the diagnostic accuracy of the ADAE algorithm for these subtypes in the "Results" section (see lines 162-165) and the newly created Table 5.

Moreover, it would be also useful to provide more information and/or hypothesis on the confounding factors that led to significantly different results in specific sub-settings (i.e. hospital 3): could it be influenced by the smaller number of the lesions? How many rare variant of melanoma or non-pigmented lesions were included?

Thank you for your helpful feedback. We have expanded our analysis to deep dive into the present outlier data (i.e., hospital 3), and updated the "Discussion" section of the revised manuscript (see lines 327-338). The dataset of hospital 3 is indeed the smallest, hence it may not be as representative. Furthermore, we have analysed the pigmented lesions in more detail (see lines 233-240, 255, and 318-320). The relative number of rare variants is comparable to other hospitals.

Reviewer #2 (Remarks to the Author):

The author conducted a prospective study to assess the effectiveness of an AI model in detecting melanoma, utilizing a homogeneous dataset that encompasses multiple melanoma subtypes sourced from various centres. The study focused on evaluating the model's generalization capabilities and benchmarking its performance against that of dermatologists. The experimental findings reveal that the AI model achieved a balanced accuracy superior to that of dermatologists, demonstrating its potential as a supportive tool for dermatologists, especially in diagnosing complex cases.

Detailed Comments:

Melanoma subtypes: the author train the Model on the public ISIC dataset with binary labels, how does it was applied to this multiple melanoma subtypes analysis? And Table 4 presents only the sensitivity, but it is not clear what the case that lesion was misclassified. To enhance clarity and

insight into the model's performance across different melanoma subtypes, it would be beneficial to include a confusion matrix in the subtype analysis.

Thank you very much for your comment. For the subtype analysis, the classification task in our study is designed binary (i.e., melanoma vs. non-melanoma), which is why there is only sensitivity presented in Table 4. We have updated the table description accordingly to enhance clarity.

While the test dataset incorporates a variety of rare melanoma classes, the number of images for some classes is notably low—for instance, certain classes are represented by merely three samples. This limited sample size could undermine the credibility of the results.

We acknowledge your concern regarding the relatively small sample size and are aware that no statistical meaningful comparisons can be performed for the concerned classes. For this reason, we have already excluded classes with a sample size <15 from the statistical analysis part in our original manuscript draft. Moreover, we have taken the opportunity to combine the concerned classes to one overarching “other” category in Table 4 (and in the newly created Table 5) of the revised manuscript to ensure the integrity of our analysis.

Classification complexity: the author evaluated the performance of AI against dermatologists by categorizing lesion complexity, which was determined based on the confidence levels of the dermatologists. However, the study lacks detailed information regarding the experience of the dermatologists involved and the methodology behind the human evaluation process. For a more comprehensive understanding, it would be beneficial to include details such as the years of experience of the dermatologist.

Thank you very much for your thoughtful comment. Unfortunately, we do not possess detailed information regarding the experience of the dermatologists, due to data privacy regulations. However, our prospective study design ensured that the involved dermatologists represent a high variance of experience levels, ranging from assistant doctors to board certified experts (as stated in the “Methods” section, lines 398-400).

Additionally, it is unclear whether the dermatologist had access to meta-information (such as age, sex, family history) during their diagnosis, which would more closely mimic a real-world diagnostic setting. Incorporating these details would provide a clearer picture of the study's context and the basis for the comparison between AI and dermatologist performance.

Thank you for your valuable feedback. The dermatologists had access to patient- and lesion-specific meta-information, as the diagnostic information were recorded as part of clinical routine within the prospective setting of our study. We have included this information in the “Results” section of the revised manuscript (see lines 127-128) to ensure clarity for our readers.

The results indicate that the AI algorithm appears to be less affected by diagnostic challenges. However, redefining lesion difficulty based on the AI's confidence level and then comparing this to human assessments could provide valuable insights. Such an approach would be instrumental in developing more effective strategies for human-AI collaboration. Additionally, the author could enhance the study by presenting examples that showcase the diagnostic preferences between the AI algorithm and dermatologists in assessing lesions.

Thank you very much for this excellent suggestion, which we believe helped us to improve the overall quality of our work. We have expanded the “Results” section of our revised manuscript by analysing the AI’s confidence level, and comparing its performance to dermatologists with respect to this feature (see lines 197-208). Moreover, we have extended Table 3 to present the analysis of the ADAE

confidence rating, updated the “Discussion” section accordingly (see line 298) and included the relative distribution of the dermatologists’ vs. the ADAE confidence scores in the supplementary materials (see Supplementary Fig. 8).

Generalization: The author asserts that the model exhibits robustness with respect to age, sex, and lesion location, yet it's important to note that the model was trained using these very variables as inputs. It may influence the conclusions drawn about the model's ability to generalize across these variables if it relies on them for evaluation. A more thorough analysis might be needed to ascertain the model's true generalizability.

Thank you for your thoughtful comment. We have added the information that relevant parameters (such as age, and lesion location), are used as input data (see lines 240-241) to enhance clarity and avoid any potential confusion.

The model's performance is significantly worse than that of dermatologist (~14% in BACC) on the data from hospital 3, however, the author did not provide sufficient analysis on this. If it is affected by age, then what is the detailed age distribution, how the error prediction distributed within the data?

Thank you for your valuable feedback. We have expanded our analysis to further investigate the concerned outlier data (i.e., hospital 3), and updated the respective paragraph within the “Discussion” section of our revised manuscript (see lines 327-338). It is also worth considering that the dataset for hospital 3 is the smallest, hence it may not be as representative.

Moreover, the author did not provide any qualitative analysis or visualization analysis for the model. e.g., umap to illustrate the feature distribution for the heterogeneous data.

Thank you very much for your suggestion. We have plotted a t-SNE representation of the dataset features, and added it as Figure 1 to the revised manuscript.

Reviewers' comments:

Reviewer #1 (Remarks to the Author):

No additional changes are required.

Reviewer #2 (Remarks to the Author):

The author conducted a prospective study to assess the effectiveness of an AI model in detecting melanoma. The study utilized a diverse dataset encompassing multiple melanoma subtypes sourced from various centers. The primary focus was to evaluate the model's generalization capabilities and benchmark its performance against that of dermatologists. The experimental findings revealed that the AI model achieved a balanced accuracy superior to that of dermatologists, demonstrating its potential as a supportive tool, especially in diagnosing complex cases.

Compared to the previous version, the author has included more details regarding the data, experiments, and results. The revised manuscript provides visualization results and additional experimental results comparing the performance of AI and humans based on the AI models' output confidence. These additions have significantly improved the clarity of the manuscript and addressed the concerns raised in the initial review. Overall, the revised manuscript has enhanced readability and effectively communicates the study's findings.

Below are minor concerns

1. All the lesions have six images, which include both polarized and non-polarized settings. Are there any results related to the different imaging settings? Is the AI model sensitive or insensitive to the imaging setting?
2. For the R-TTA (Test-Time Augmentation), does it mean that all six images are subjected to TTA separately, and then the results are averaged?

Response to Reviewers

We sincerely thank the reviewers for their diligent assessment of our work and for their insightful comments and recommendations.

Reviewer #1 (Remarks to the Author):

No additional changes are required.

Thank you very much.

Reviewer #2 (Remarks to the Author):

The author conducted a prospective study to assess the effectiveness of an AI model in detecting melanoma. The study utilized a diverse dataset encompassing multiple melanoma subtypes sourced from various centers. The primary focus was to evaluate the model's generalization capabilities and benchmark its performance against that of dermatologists. The experimental findings revealed that the AI model achieved a balanced accuracy superior to that of dermatologists, demonstrating its potential as a supportive tool, especially in diagnosing complex cases.

Compared to the previous version, the author has included more details regarding the data, experiments, and results. The revised manuscript provides visualization results and additional experimental results comparing the performance of AI and humans based on the AI models' output confidence. These additions have significantly improved the clarity of the manuscript and addressed the concerns raised in the initial review. Overall, the revised manuscript has enhanced readability and effectively communicates the study's findings.

Below are minor concerns:

All the lesions have six images, which include both polarized and non-polarized settings. Are there any results related to the different imaging settings? Is the AI model sensitive or insensitive to the imaging setting?

Thank you very much for your comment. Unfortunately, there are no specific results related to those settings, as this was not the focus of this work.

For the R-TTA (Test-Time Augmentation), does it mean that all six images are subjected to TTA separately, and then the results are averaged?

Thank you very much for your comment. The TTA happens using the six images, that is they are predicted separately, and then their results are averaged to one final prediction for the respective lesion. We have expanded our description in the revised manuscript to enhance clarity (see ll. 131-135).

REVIEWERS' COMMENTS:

Reviewer #2 (Remarks to the Author):

The authors have addressed all my previous concerns, the current version is clear, therefore, I have no further comments.